# Graph Mixture of Experts: Learning on Large-Scale Graphs with Explicit Diversity Modeling

**Haotao Wang**[1]*, **Ziyu Jiang**[2]*, **Yuning You**[2], **Yan Han**[1], **Gaowen Liu**[3], **Jayanth Srinivasa**[3]
**Ramana Rao Kompella**[3], **Zhangyang Wang**[1]

[1]University of Texas at Austin      Texas A&M University[2]      Cisco Systems[3]

{htwang, yh9442, atlaswang}@utexas.edu, {jiangziyu, yuning.you}@tamu.edu,
{gaoliu, jasriniv, rkompell}@cisco.com

## Abstract

Graph neural networks (GNNs) have found extensive applications in learning from graph data. However, real-world graphs often possess diverse structures and comprise nodes and edges of varying types. To bolster the generalization capacity of GNNs, it has become customary to augment training graph structures through techniques like graph augmentations and large-scale pre-training on a wider array of graphs. Balancing this diversity while avoiding increased computational costs and the notorious trainability issues of GNNs is crucial. This study introduces the concept of Mixture-of-Experts (MoE) to GNNs, with the aim of augmenting their capacity to adapt to a diverse range of training graph structures, without incurring explosive computational overhead. The proposed *Graph Mixture of Experts* (**GMoE**) model empowers individual nodes in the graph to dynamically and adaptively select more general *information aggregation experts*. These experts are trained to capture distinct subgroups of graph structures and to incorporate information with varying hop sizes, where those with larger hop sizes specialize in gathering information over longer distances. The effectiveness of GMoE is validated through a series of experiments on a diverse set of tasks, including graph, node, and link prediction, using the OGB benchmark. Notably, it enhances ROC-AUC by $1.81\%$ in ogbg-molhiv and by $1.40\%$ in ogbg-molbbbp, when compared to the non-MoE baselines. Our code is publicly available at `https://github.com/VITA-Group/Graph-Mixture-of-Experts`.

## 1   Introduction

Graph learning has found extensive use in various real-world applications, including recommendation systems [1], traffic prediction [2], and molecular property prediction [3]. Real-world graph data typically exhibit diverse graph structures and heterogeneous nodes and edges. In graph-based recommendation systems, for instance, a node can represent a product or a customer, while an edge can indicate different interactions such as view, like, or purchase. Similarly in biochemistry tasks, datasets can comprise molecules with various biochemistry properties and thherefore various graph structures. Moreover, purposefully increasing the diversity of graph data structures in training sets has become a crucial aspect of GNN training. Techniques such as graph data augmentations [4, 5] and large-scale pre-training on diverse graphs [6, 7, 8, 9, 10] have been widely adopted to allow GNNs for extracting more robust and generalizable features. Meanwhile, many real-world GNN applications, such as recommendation systems and molecule virtual screening, usually involve processing a vast number of candidate samples and therefore demand computational efficiency. That invites the key

---

*The first two authors contributed equally.

37th Conference on Neural Information Processing Systems (NeurIPS 2023).

question: *Can one effectively scale a GNN model's **capacity** to leverage larger-scale, more diverse graph data, without compromising its **inference efficiency**?*

A common limitation of many GNN architectures is that they are essentially "homogeneous" across the whole graph, i.e., forcing all nodes to share the same aggregation mechanism, regardless of the differences in their node features or neighborhood information.[2] That might be suboptimal when training on diverse graph structures, e.g, when some nodes may require information aggregated over longer ranges while others prefer shorter-range local information. Our solution is the proposal of a novel GNN architecture dubbed *Graph Mixture of Experts* (**GMoE**). It comprises multiple "experts" at each layer, with each expert being an independent message-passing function with its own trainable parameters. The idea establishes a new base to address the diversity challenges residing in graph data.

Throughout the training process, GMoE is designed to intelligently select aggregation experts tailored to each node. Consequently, nodes with similar neighborhood information are guided towards the same aggregation experts. This fosters specialization within each GMoE expert, focusing on specific subsets of training samples with akin neighborhood patterns, regardless of range or aggregation levels. In order to harness the full spectrum of diversity, GMoE also incorporates aggregation experts with distinct inductive biases. For example, each GMoE layer is equipped with aggregation experts of varying hop sizes. Those with larger hop sizes cater to nodes requiring information from more extended ranges, while the opposite holds true for those with smaller hop sizes.

We have rigorously validated GMoE's effectiveness through a range of comprehensive molecular property prediction tasks, underscoring our commitment to deliberate diversity modeling. Moreover, our analysis demonstrates that GMoE surpasses other GNN models in terms of *inference efficiency*, even when they possess similar-sized parameters, thanks to the dynamic expert selection. This efficiency proves crucial in real-world scenarios, such as virtual screening in libraries of trillion-scale magnitude or beyond. The potency of our approach is corroborated by extensive experiments on ten graph learning datasets within the OGB benchmark. For instance, GMoE enhances the ROC-AUC by $1.81\%$ on ogbg-molhiv, $1.40\%$ on ogbg-molbbbp, $0.95\%$ on ogbn-proteins, and boosts Hits@20 score by $0.89\%$ on ogbl-ddi, when compared to the single-expert baseline. To gain deeper insights into our method, we conduct additional ablation studies and comprehensive analyses.

## 2 Related Work

**Graph Neural Networks** Graph neural networks (GNNs) [11, 12, 13] have emerged as a powerful approach for learning graph representations. Variants of GNNs have been proposed [11, 12, 13], achieving state-of-the-art performance in different graph tasks. Under the message passing framework [14], graph convolutional network (GCN) adopts mean pooling to aggregate the neighborhood and updates embeddings recursively [11]; GraphSAGE [15] adopts sampling and aggregation schemes to eliminate the inductive bias in degree; and graph attention network (GAT) utilizes the learnable attention weights to adaptively aggregate. To capture long-range dependencies in disassortative graphs, Geom-GCN devises a geometric aggregation scheme [16] to enhance the convolution, benefiting from a continuous space underlying the graph. Lately, Graphormer [17] proposes a novel graph transformer model that utilizes attention mechanisms to capture the structural information.

**Mixture of Experts** The concept of Mixture of Experts (MoE) [18] has a long history, tracing its origins back to earlier work s[19, 20, 21]. Recently, spurred by advancements in large language models, sparse MoE [22, 23, 24, 25, 26] has re-gained prominence. This variant selectively activates only a small subset of experts for each input, significantly enhancing efficiency and enabling the development of colossal models with trillions of parameters. This breakthrough has revolutionized the learning process, particularly on vast language datasets [25, 27]. Subsequent studies have further refined the stability and efficiency of sparse MoE [28, 29]. The remarkable success of sparse MoE in the realm of language has spurred its adoption in diverse domains, including vision [30, 31], multi-modal [32], and multi-task learning [33, 34, 35].

**MoE for GNNs** In the domain of graph analysis, prior research has assembled knowledge from multiple ranges by combining various GNNs with different scopes [36, 37], akin to a fixed-weight

---

[2]There are exceptions. For example, GAT and GraphTransformer adaptively learn the aggregation function for each node. This paper focuses on discussing whether GMoE can improve common homogeneous GNNs such as GCN and GIN. Extending GMoE over heterogeneous GNNs is left for future research.

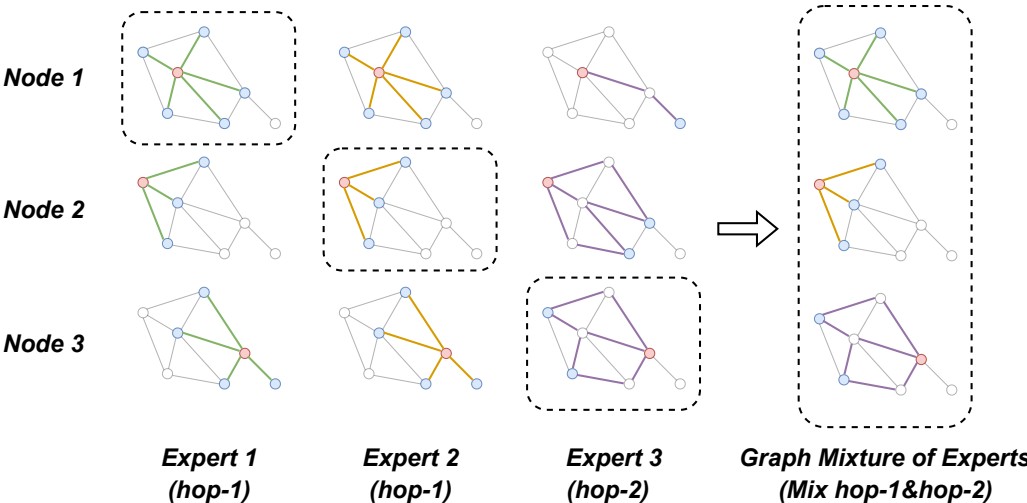

Figure 1: Each row represents the aggregation of a single node, and each column corresponds to a different network or sub-module. Blue dots (•) denote the input features passed to the red dots (•) via the colorful edges. On the left, we demonstrate two hop-1 experts with distinct weights, along with one hop-2 expert. On the right, GMoE is depicted. In this instance, the proposed GMoE selectively chooses one expert for each node while masking the others. Best viewed in color.

Mixture of Experts (MoE). Pioneering efforts [38, 39] have also investigated the application of MoE to address the well-known issue of imbalance and to develop unbiased classification or generalization algorithms. However, none of these approaches harnessed the potential advantages of sparsity and adaptivity. Recent work [40] introduced the use of a mixture of experts for molecule property prediction. They employed a GNN as a feature extractor and applied a mixture of experts, where each expert is a linear classifier, on top of the extracted features for graph classification. In contrast, each layer of GMoE constitutes a mixture of experts, with each expert being a GCN/GIN layer featuring different aggregation step sizes. Another distinction from [40] lies in their utilization of domain-specific knowledge (specifically, molecule topology) for expert routing, whereas our approach is designed to operate on general graphs without relying on domain-specific assumptions. One more concurrent study by [41] employs MoE to achieve fairness in predictions for GNNs.

Our study takes a significant stride forward by introducing sparse MoE to scale graph neural networks in an end-to-end fashion, enabling efficient learning on datasets featuring diverse graph structures. We incorporate experts with varying scopes, allowing the gating function to dynamically select neighbors with the desired range.

**Adapting Deep Architectures for Training Data Diversity** Several prior studies have delved into enhancing the capacity of generic deep neural networks to effectively leverage a wide array of training samples without incurring additional inference costs. For instance, [42] suggested employing two distinct batch normalization (BN) layers for randomly and adversarially augmented training samples, based on the observation that these two sets of augmented samples originate from different distributions. Building upon this concept, [43] extended it by introducing an auxiliary instance normalization layer to further reduce the heterogeneity of input features before reaching the BN layers. More recently, [44] demonstrated that normalizer-free convolutional networks (CNNs) [45, 46] exhibit significantly greater capability in accommodating diverse training sets compared to conventional BN-based CNNs. However, these prior works have primarily concentrated on devising improved normalization strategies for CNNs, while Graph Neural Networks (GNNs) often do not rely on (batch) normalization as heavily.

# 3 Method

**Preliminaries: Graph Neural Networks**    Taking the classical Graph Convolutional Network (GCN) [11] as an example, the propagation mechanism can be formulated as

$$h_i' = \sigma \left( \sum_{j \in N_i} \frac{1}{\sqrt{|N_i||N_j|}} h_j W^{(i)} \right),$$
(1)

where $W^{(i)} \in \mathbb{R}^{s \times s}$ is a trainable weight and $\sigma$ is an element-wise non-linear activation function. $h_j \in \mathbb{R}^{b \times s}$ denotes the input feature of $j$ th node while $h_i' \in \mathbb{R}^{b \times s}$ is its output feature in $i$ th node. $b$ and $s$ are batch size and hidden feature size, respectively. $N_i$ denotes the collection of neighbors for $i$ th node including self-connection. The output feature is normalized by $\frac{1}{\sqrt{|N_i||N_j|}}$. A canonical GCN layer only aggregates the information from immediately adjacent neighbors (hop-1).

## 3.1 Graph Mixture of Experts

The general framework of GMoE is outlined in Figure 1. The GMoE layer comprises multiple experts, each utilizing either the hop-1 or hop-2 aggregation function. To determine which experts to use for a given node, a gating function is employed. This allows for similar nodes to be assigned to the same experts when learning with diverse graph structures, thereby enabling each expert to specialize in a particular structure type. By doing so, the model can more effectively capture diverse graph structures present within the training set. The GMoE layer's adaptive selection between the hop-1 and hop-2 experts enables the model to dynamically capture short-range or long-range information aggregation for each node. Formally, a GMoE layer can be written as:

$$h_i' = \sigma \left( \sum_{o=1}^{m} \sum_{j \in N_i} G(h_i)_o E_o \left( h_j, e_{ij}, W \right) + \sum_{o=m}^{n} \sum_{j \in N_i^2} G(h_i)_o E_o \left( h_j, e_{ij}, W \right) \right),$$
(2)

where $m$ and $n$ denote the hop-1 and total experts number, respectively. Hench the number of hop-2 experts is $n - m$. $E_o$ and $e_{ij}$ denote the message function and edge feature between $i$ th and $j$ th nodes, respectively. It can represent multiple types of message-passing functions such as one employed by GCN [47] or GIN [13]. $G$ is the gating function that generates multiple decision scores with the input of $h_i$ while $G(h_i)_o$ denotes the $o$ th item in the output vector of $G$. We employ the noisy top-$k$ gating design for $G$ following [22], which can be formalized with

$$G(h_i) = \text{Softmax}(\text{TopK}(Q(h_i), k)),$$
(3)

$$Q(h_i) = h_i W_g + \epsilon \cdot \text{Softplus}(h_i W_n),$$
(4)

where $k$ denotes the number of selected experts. $\epsilon \in \mathcal{N}(0,1)$ denotes standard Gaussian noise. $W_g \in \mathbb{R}^{s \times n}$ and $W_n \in \mathbb{R}^{s \times n}$ are learnable weights that control clean and noisy scores, respectively.

The proposed GMoE layer can be applied to many GNN backbones such as GIN [13] or GCN [11]. In practice, we replace every layer of the backbone with its corresponding GMoE layer. For simplicity, we name the resultant network GMoE-GCN or GMoE-GIN (for GCN and GIN, respectively).

**Additional Loss Functions to Mitigate GMoE Collapse**    Nonetheless, if this model is trained solely using the expectation-maximization loss, it may succumb to a trivial solution wherein only a single group of experts is consistently selected. This arises due to the self-reinforcing nature of the imbalance: the chosen experts can proliferate at a much faster rate than others, leading to their increased frequency of selection. To mitigate this, we implement two additional loss functions to prevent such collapse [22]. The first one is importance loss:

$$\text{Importance}(H) = \sum_{h_i \in H, g \in G(h_i)} g, \quad L_{\text{importance}}(H) = CV(\text{Importance}(H))^2,$$
(5)

where the importance score $\text{Importance}(H)$ is defined as the sum of each node's gate value $g$ across the whole batch. $CV$ represents the coefficient of variation. The importance loss $L_{\text{importance}}(H)$ hence measures the variation of importance scores, enforcing all experts to be "similarly important".

While the importance score enforces equal scoring among the experts, there may still be disparities in the load assigned to different experts. For instance, one expert could receive a few high scores, while another might be selected by many more nodes ye t all with lower scores. This situation can potentially lead to memory or efficiency issues, particularly on distributed hardware setups. To address this, we introduce an additional load-balanced loss to encourage a more even selection probability per expert. Specifically, $G(h_i) \neq 0$ if and only if $Q(h_i)_o$ is greater than the $k$-th largest element of $Q(h_i)$ excluding itself. Consequently, the probability of $G(h_i) \neq 0$ can be formulated as:

$$P(h_i, o) = Pr(Q(h_i)_o > \text{kth\_ex}(Q(h_i), k, o)), \tag{6}$$

where $kth\_ex()$ denotes the $k$-th largest element excluding itself. $P(h_i, o)$ can be simplified as

$$P(h_i, o) = \Phi\left(\frac{h_i W_g - \text{kth\_ex}(Q(h_i), k, o)}{\text{Softplus}(h_i W_n)}\right), \tag{7}$$

where $\Phi$ is the CDF of standard normal distribution. The load is then defined as ($p$ is the node-wise probability in the batch):

$$L_{\text{load}}(H) = CV(\sum\nolimits_{h_i \in H, p \in P(h_i, o)} p)^2. \tag{8}$$

The final loss employs both the task-specific loss and two load-balance losses, leading to the overall optimization target ($\lambda$ is a hand-tuned scaling factor):

$$L = L_{EM} + \lambda(L_{\text{load}}(H) + L_{\text{importance}}(H)), \tag{9}$$

where $L_{EM}$ denotes the task-specific MoE expectation-maximizing loss.

**Pre-training GMoE**    We further discover that GMoE could be combined with and strengthened by the self-supeervised graph pre-training techniques. We employ GraphMAE [10] as the self-supervised pre-training technique, defined as

$$\mathcal{L}(H, M) = D\left(d\left(f\left(M \cdot H\right)\right), H\right), \tag{10}$$

where $f$ and $d$ denote the encoder and the decoder networks, and $M$ represents the mask for the input graph $H$. $f$ and $d$ collaborative conduct the reconstruction task from the corrupted input, whose quality is measured by the distance metric $D$. We later will experimentally demonstrate and compare GMoE performance with and without pre-training.

## 3.2   Computational Complexity Analysis

We show that GMoE-GNN brings negligible overhead on the inference cost compared with its GNN counterpart. We measure computational cost. using the number of floating point operations (FLOPs).

The computation cost of a GMoE layer can be defined as

$$C_{GMoE} = \sum_{h_i \in H} \mathcal{F}\left[\sum_{o=1}^{m} G(h_i)_o \sum_{j \in N_i} E_o\left(h_j, e_{ij}, W\right) + \sum_{o=m}^{n} G(h_i)_o \sum_{j \in N_i^2} E_o\left(h_j, e_{ij}, W\right)\right], \tag{11}$$

where $\mathcal{F}$ maps functions to its flops number. $C_{GMoE}$ denotes the computation cost of the whole layer in GMoE-GCN. Given there exists an efficient algorithm that can solve hop-1 and hop-2 functions with matching computational complexity [36], we can further simplify $C_{GMoE}$ as

$$C_{GMoE} = \sum_{h_i \in H} \sum_{j \in N_i} C \sum_{o=1}^{n} \mathbf{1}(G(h_i)_o), \tag{12}$$

$$C = \mathcal{F}\left[E_o\left(h_j, e_{ij}, W\right)\right], \quad \mathbf{1}(G(h_i)_o) = \begin{cases} 0 & \text{if } G(h_i)_o = 0, \\ 1 & \text{otherwise.} \end{cases} \tag{13}$$

Given $\sum_{o=1}^{n} \mathbf{1}(G(h_i)_o) = k$, $C_{GMoE}$ can be further simplified to

$$C_{GMoE} = k \sum_{h_i \in H} \sum_{j \in N_i} C. \tag{14}$$

Table 1: Performance Summary for Graph Classification Tasks. The table header lists the dataset names and corresponding evaluation metrics. Mean and standard deviation values from ten random runs are presented. The most outstanding results are highlighted in bold. The improvements achieved by GMoE are indicated in parentheses. ROC-AUC scores are reported in percentage.

| Model | ogbg-molbbbp | ogbg-molhiv | ogbg-moltoxcast | ogbg-moltox21 |
| --- | --- | --- | --- | --- |
| | | ROC-AUC ($\uparrow$) | | |
| GCN | $68.87 \pm 1.51$ | $76.06 \pm 0.97$ | $63.54 \pm 0.42$ | $75.29 \pm 0.69$ |
| GMoE-GCN | $\mathbf{70.28} \pm 1.36$ (+1.41) | $\mathbf{77.87} \pm 1.03$ (+1.81) | $\mathbf{64.12} \pm 0.61$ (+0.58) | $\mathbf{75.45} \pm 0.58$ (+0.16) |

Table 2: Summary of Graph Regression Task Results. The table header displays the dataset names and the corresponding evaluation metrics. Mean and standard deviation values from ten random runs are presented. The most outstanding results are highlighted in bold. GMoE's performance improvements are indicated in parentheses. All evaluation metrics are reported in percentage.

| Model | ogbg-molesol | ogbg-molfreesolv |
| --- | --- | --- |
| | RMSE ($\downarrow$) | |
| GCN | $1.114 \pm 0.036$ | $2.640 \pm 0.239$ |
| GMoE-GCN | $\mathbf{1.087} \pm 0.043$ (-0.027) | $\mathbf{2.500} \pm 0.193$ (-0.140) |

Here, $C$ is the computation cost of a single message passing in GMoE-GCN. Denote the computation cost of a single GCN message passing as $C_0$, and the total computational cost in the whole layer as $C_{GCN}$. By setting $C = \frac{C_0}{k}$ in GMoE-GCN, we have $C_{GMoE} = \sum_{h_i \in H} \sum_{j \in N_i} C_0 = C_{GCN}$.

In traditional GMoE-GCN or GMoE-GIN, the adjustment of $C$ can be easily realized by controlling the hidden feature dimension size $s$. For instance, GMoE-GCN and GMoE-GIN with hidden dimension size of $s = \frac{s_0}{\sqrt{k}}$ can have similar FLOPs with its corresponding GCN and GIN with dimension size of $s_0$. The computation cost of gating functions in GMoE is meanwhile negligible compared to the cost of selected experts, since both $W_g \in \mathbb{R}^{n \times s}$ and $W_n \in \mathbb{R}^{n \times s}$ is in a much smaller dimension than $W^{(i)} \in \mathbb{R}^{s \times s}$ given $n \ll s$.

In practice, on our NVIDIA A6000 GPU, the inference times for $10,000$ samples are $30.2 \pm 10.6ms$ for GCN-MoE and $36.3 \pm 17.2ms$ for GCN. The small variances in GPU clock times align with their nearly identical theoretical FLOPs.

## 4   Experimental Results

In this section, we first describe the detailed settings in Section 4.1. We then show our main results on graph learning in Section 4.2. Ablation studies and analysis are provided in Section 4.3.

### 4.1   Experimental Settings

**Datasets and Evaluation Metrics**   We conduct experiments on ten graph datasets in the OGB benchmark [48], including graph-level (i.e., ogbg-bbbp, ogbg-hiv, ogbg-moltoxcast, ogbg-moltox21, ogbg-molesol, and ogbg-freesolv), node-level (i.e., ogbn-protein, ogbn-arxiv), and link-level prediction (i.e., ogbl-ddi, ogbl-ppa) tasks. Following [48], we use ROC-AUC (i.e., area under the receiver operating characteristic curve) as the evaluation metric on ogbg-bbbp, ogbg-hiv, ogbg-moltoxcast, ogbg-moltox21, and ogbn-protein; RMSE (i.e., root mean squared error) on ogbg-molesol, and ogbg-freesolv; classification accuracy (Acc) on ogbn-arxiv; Hits@100 score on ogbl-ppa; Hits@20 score on ogbl-ddi.

**Model Architectures and Training Details**   We use the GCN [11] and GIN [13] provided by OGB benchmark [48] as the baseline models. All model settings (e.g., number of layers, hidden feature dimensions, etc.) and training hyper-parameters (e.g., learning rates, training epochs, batch size, etc.)

Table 3: Performance Comparison for Graph Classification Tasks using GIN with and without Pre-training. We employ GraphMAE [10] for pre-training. The table lists four datasets along with the evaluation metric. The final column displays the average performance across these datasets. Mean and standard deviation values from ten random runs are provided. The most notable results are highlighted in bold.

| Model | ogbg-molbbbp | ogbg-molhiv | ogbg-moltoxcast ROC-AUC (↑) | ogbg-moltox21 | Average |
|---|---|---|---|---|---|
| GIN | $65.5 \pm 1.8$ | $75.4 \pm 1.5$ | $63.3 \pm 1.5$ | $74.3 \pm 0.5$ | 69.63 |
| GMoE-GIN | $\mathbf{66.93} \pm 1.72$ (+1.43) | $\mathbf{76.14} \pm 1.03$ (+0.74) | $62.86 \pm 0.37$ (-0.44) | $\mathbf{74.76} \pm 0.66$ (+0.46) | $\mathbf{70.17}$ (+0.54) |
| GIN+Pretrain | $\mathbf{69.94} \pm 0.92$ | $76.1 \pm 0.8$ | $62.96 \pm 0.55$ | $73.85 \pm 0.64$ | 70.71 |
| GMoE-GIN+Pretrain | $68.62 \pm 1.02$ (-1.32) | $\mathbf{76.9} \pm 0.9$ (+0.8) | $\mathbf{64.48} \pm 0.50$ (+1.18) | $\mathbf{75.25} \pm 0.78$ (+1.40) | $\mathbf{71.31}$ (+0.6) |

Table 4: Node Prediction Task Results. The table header displays the dataset names and the corresponding evaluation metrics, both reported in percentage. Mean and standard deviation values from ten random runs are presented. The most notable results are highlighted in bold, with GMoE's performance gains indicated in parentheses

| Model | ogbn-protein ROC-AUC(↑) | ogbn-arxiv Acc(↑) |
|---|---|---|
| GCN | $73.53 \pm 0.56$ | $71.74 \pm 0.29$ |
| GMoE-GCN | $\mathbf{74.48} \pm 0.58$ (+0.95) | $\mathbf{71.88} \pm 0.32$ (+0.14) |

are identical as those in [48]. We show the performance gains brought by their GMoE counterparts: GMoE-GCN and GMoE-GIN. For GMoE models, as described in Section 3, we select $k$ experts out of a total of $n$ experts for each node, where $m$ out of $n$ experts are hop-1 aggregation functions and the rest $n - m$ are hop-2 aggregation functions. All three hyper-parameters $n, m, k$, together with the loss trade-off weight $\lambda$ in Eq. (9), are tuned by grid searching: $n \in \{4, 8\}, m \in \{0, n/2, n\}, k \in \{1, 2, 4\}$, and $\lambda \in \{0.1, 1\}$. The hidden feature dimension would be adjusted with $k$ (Section 3.2) to ensure the same flops for all comparisons. The hyper-parameter values achieving the best performance on validation sets are selected to report results on test sets, following the routine in [48]. All other training hyper-parameters on GMoE (e.g., batch size, learning rate, training epochs) are kept the same as those used on the single-expert baselines. All experiments are run for ten times with different random seeds, and we report the mean and deviation of the results following [48].

**Pre-training Settings** Following the transfer learning setting of [49, 6, 8, 10], we pre-train the models on a subset of the ZINC15 dataset [50] containing 2 million unlabeled molecule graphs. For training hyperparameters, we employ a batch size of 1024 to accelerate the training on the large pre-train dataset for both baselines and the proposed method. We follow [10] employing GIN [13] as the backbone, 0.001 as the learning rate, adam as the optimizer, 0 as the weight decay, 100 as the training epochs number, and 0.25 as the masking ratio.

## 4.2 Main Results

Our evaluation primarily centers around comparing GMoE-GCN with the baseline single-expert GCN using six graph property prediction datasets. These datasets encompass four graph classification tasks and two regression tasks within a supervised learning framework. In all cases, models are trained from scratch, utilizing only the labeled samples in the training set. The classification and regression results are outlined in Table 1 and 2, respectively.

Table 5: Link Prediction Task Results. The table header provides the dataset names and the respective evaluation metrics, both expressed in percentage. Mean and standard deviation values from ten random runs are presented. The most noteworthy results are highlighted in bold, with GMoE's performance improvements indicated in parentheses.

| Model | ogbl-ppa Hits@100($\uparrow$) | ogbl-ddi Hits@20($\uparrow$) |
|---|---|---|
| GCN | $18.67 \pm 1.32$ | $37.07 \pm 0.051$ |
| GMoE-GCN | $\mathbf{19.25} \pm 1.67$ (+0.58) | $\mathbf{37.96} \pm 0.082$ (+0.89) |

Table 6: Graph Property Prediction Results employing GMoE-GCN with Varied Hyper-Parameter Configurations. The third column displays the average number of nodes within graphs across different datasets. Mean and standard deviation values from ten random runs are provided. The most noteworthy outcomes are emphasized in bold. ROC-AUC scores are expressed in percentage.

| Dataset | Metric | Average #Nodes | $n = m = 4$ $k = 4$ | $n = m = 4$ $k = 1$ | $n = m = 8$ $k = 4$ | $n = 8, m = 0$ $k = 4$ |
|---|---|---|---|---|---|---|
| ogbg-molfreesolv | RMSE ($\downarrow$) | 8.7 | $2.856 \pm 0.118$ | $\mathbf{2.500} \pm 0.193$ | $2.532 \pm 0.231$ | $2.873 \pm 0.251$ |
| ogbg-molhiv | ROC-AUC ($\uparrow$) | 25.5 | $76.67 \pm 1.25$ | $76.41 \pm 1.72$ | $77.53 \pm 1.42$ | $\mathbf{77.87} \pm 1.03$ |

It is worth noting that GMoE-GCN consistently outperforms the baseline across all six datasets. Notably, there are substantial improvements in ROC-AUC, with increases of $1.81\%$ on the molhiv dataset and $1.41\%$ on the molbbp dataset. While these enhancements may seem modest, they represent significant progress. Additionally, lifts of $1.40\%$ on ogbg-moltox21, $1.43\%$ on ogbg-molbbbq, and $1.18\%$ on ogbg-moltoxcast have been observed. The uniformity of these improvements across diverse tasks and datasets underscores the reliability and effectiveness of GMoE-GCN.

Leveraging large-scale pretraining on auxiliary unlabeled data has notably enhanced the generalization capabilities of GNNs even further. Expanding on this, our GMoE consistently improves performance when integrated with large-scale pretraining methods. Following the methodology outlined in [10], we employ GIN [13] as the baseline model for comparison with GMoE-GIN. As illustrated in Table 3, GMoE-GIN outperforms GIN on 3 out of 4 datasets, enhancing the average performance by $0.54\%$, even without pretraining. This underscores the versatility of GMoE in enhancing various model architectures. When coupled with the pretraining, GMoE-GIN further widens the performance gap with the GIN baseline, achieving a slightly more pronounced improvement margin of $0.6\%$

Additionally, GMoE showcases its potential in node and link prediction tasks. Experiments conducted on ogbn-protein and ogbn-arxiv for node prediction, as well as ogbl-ddi and ogbl-ppa for link prediction, further validate this observation. The results, detailed in Table 4 and Table 5, demonstrate GMoE-GCN's superiority over the single-expert GCN. Notably, it enhances performance metrics like ROC-AUC by $0.95\%$ and Hits@20 by $0.89\%$ on the ogbn-protein and ogbl-ddi datasets.

## 4.3 Ablation Study and Analysis

**Observation 1: Larger graphs prefer larger hop sizes** We conducted an ablation study on two prominent molecule datasets, namely ogbg-molhiv and ogbg-molfreesolv, which exhibit the largest and smallest average graph sizes, respectively, among all molecular datasets in the OGB benchmark. As illustrated in the third column of Table 6, the average size of molecular graphs in ogbg-molhiv is approximately three times greater than that of the ogbg-molfreesolv dataset. Remarkably, on ogbg-molfreesolv, if all experts utilize hop-2 aggregation functions (i.e., when $m = 0$), the performance substantially lags behind the scenario where all experts employ hop-1 aggregations (i.e., when $m = n$). In contrast, on ogbg-molhiv, characterized by significantly larger graphs, leveraging all hop-2 experts (i.e., when $m = 0$) yields superior performance compared to employing all hop-1 experts (i.e., when $m = n$). This observation suggests that larger graphs exhibit a preference for aggregation experts with greater hop sizes. This alignment with intuition stems from the notion

Table 7: Results for graph property prediction using GMoE-GCN with different scaling ratios $\lambda$ for the loading balance losses on moltox21. Mean and standard deviation over ten random runs are reported. The best results are shown in bold. ROC-AUC is reported in percentage.

| $\lambda$ | 0 | 0.1 | 1 |
|---|---|---|---|
| ROC-AUC | 72.87$\pm$1.04 | **75.45**$\pm$0.58 | 75.27$\pm$0.33 |

that larger molecular graphs may necessitate more extensive long-range aggregation information in contrast to their smaller counterparts.

**Observation 2: Sparse expert selection helps not only efficiency, but also generalization** As depicted in Table 6, the optimal performance on both datasets is attained with sparse MoEs (i.e., when $k < n$), as opposed to a full dense model (i.e., when $k = n$). This seemingly counter-intuitive finding shows another benefit of sparsity besides significantly reducing inference computational cost: that is, sparsity is also promising to improve model *generalization*, particularly when dealing with multi-domain data, as it allows a group of experts to learn collaboratively and generalize to unseen domains compositionally. Our finding also echoes prior work, e.g., [51], which empirically shows sparse MoEs are strong domain generalizable learners.

**Observation 3: GMoE demonstrates performance gains, though converging over more epochs** We compared the convergence curves of GMoE-GCN and the single-expert GCN, illustrated in Figure 2. Specifically, we plotted the validation and test ROC-AUC at different epochs while training on the protein dataset. As observed, the performance of GMoE-GCN reaches a plateau later than that of GCN. However, GMoE-GCN eventually achieves a substantial performance improvement after a sufficient number of training epochs.

We provide an explanation for this phenomenon below. In the initial stages of training, each expert in GMoE has been updated for $k$ times fewer iterations compared to the single expert GNN. Consequently, all experts in GMoE are relatively weak, leading to GMoE's inferior performance compared to GCN during this early phase. However, after GMoE has undergone enough epochs of training, all experts have received ample updates, enabling them to effectively model their respective subgroups of data. As a result, the performance of GMoE surpasses that of the single-expert GCN.

Figure 2: Convergence curve of GMoE-GCN and GCN when trained on the ogb-protein dataset. The solid curves and the shaded areas are the mean and standard deviations of ROC-AUC calculated over ten random runs.

**Observation 4: Load balancing is essential** We conducted an investigation into the scaling factor for the load balancing loss, denoted by $\lambda$. As shown in Table 7, utilizing $\lambda = 0.1$ resulted in significantly improved performance, exceeding 2.58% in terms of ROC-AUC compared to when $\lambda = 0$. This underscores the crucial role of implementing load-balancing losses. Conversely, the choice of $\lambda$ exhibits less sensitivity; opting for $\lambda = 1$ would yield a similar performance of 75.27%.

**Observation 5: GMoE improves other state-of-the-art GNN methods** Finally, we explore whether our proposed GMoE can yield an improvement in the performance of other state-of-the-art models listed on the OGB leaderboard. We selected Neural FingerPrints [52] as our baseline. As of the time of this submission, Neural FingerPrints holds the fifth position in the ogbg-molhiv benchmark. It is noteworthy that this methodology has gained significant acclaim, as it serves as the foundation for the top four current methods [53, 54, 55, 56].

After integrating the GMoE framework into the Neural FingerPrints architecture, we achieved an accuracy of 82.72% ± 0.53%, while maintaining the same computational resource requirements. This performance outperforms Neural FingerPrints by a margin of 0.4%, underlining the broad and consistent adaptability of the GMoE framework.

## 5 Conclusion

In this work, we propose the Graph Mixture of Experts (GMoE) model, aiming at addressing the challenges posed by diverse graph structures. By incorporating multiple experts at each layer, each equipped with its own trainable parameters, GMoE introduces a novel approach to modeling graph data. Through intelligent expert selection during training, GMoE ensures nodes with similar neighborhood information are directed towards the same aggregation experts, promoting specialization within each expert for specific subsets of training samples. Additionally, the inclusion of aggregation experts with distinct inductive biases further enhances GMoE's adaptability to different graph structures. Our extensive experimentation and analysis demonstrate GMoE's notable accuracy-efficiency trade-off improvements over baselines. These advancements hold great promise for real-world applications, particularly in scenarios requiring the efficient processing of vast amounts of candidate samples.

**Limitations:** As an empirical solution to enhance the capability of GNNs to encode diverse graph data, we primarily assess the effectiveness of the proposed method using empirical experimental results. It is important to note that our initial comparison primarily focused on fundamental backbones such as GCN/GIN to elucidate our concept. Recognizing the crucial importance of a comprehensive evaluation, we are actively broadening our comparative scope to include state-of-the-art models derived from GCN and other MoE models. Future iterations will thus incorporate a more exhaustive evaluation. Moreover, given the prevalence of GCN in semi-supervised learning, there is potential in exploring the benefits of our proposed approach in such tasks – an avenue we are eager to explore further. Another open question stemming from this work is whether we can apply GMoE on GNNs with heterogeneous aggregation mechanisms, such as GAT and GraphTransformer, which may potentially further improve the performance on diverse graph data.

## Acknowledgement

The work is sponsored by a Cisco Research Grant (UTAUS-FA00002062).

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
