# OpenReview forum: "Graph Mixture of Experts: Learning on Large-Scale Graphs with Explicit Diversity Modeling"
_NeurIPS.cc/2023/Conference — NeurIPS 2023 poster_

### Official Review · Reviewer_NqGe · 2023-07-01

**Soundness:** 3 good
**Presentation:** 3 good
**Contribution:** 3 good
**Rating:** 6
**Confidence:** 4

**Summary:**

This paper considers the problem of graph neural networks. After discussing the drawback of existing methods, this paper presents a new model, called  Graph Mixture of Expert (GMoE), which can dynamically select its own optimal information aggregation experts. Extensive experiments conducted on several graph datasets show the benefit of the proposed method.

**Strengths:**

+ The authors have carefully reviewed the drawback of existing graph models and then propose a novel and effective model, called Graph Mixture of Expert (GMoE). The GMoE enables the model to dynamically select its own optimal information aggregation experts.

+ The paper is well-written and easy to follow. The reviewer can easily understand the motivation of this paper and the details of the proposed method.

+ Experiments have shown the benefit of the proposed method over the traditional GCN.

**Weaknesses:**

- Although the authors have shown the benefit of the proposed method over GCN, the improvement seems somewhat limited. For example, in several tables, the improvement over GCN is only 0.*%.

- It seems that the comparison to other methods is limited. The authors have only compared to the traditional GCN. However, the comparison to other state-of-the-art models modified based on GCN is missed. In addition, it is very important to compare with other MoE models.

- As GCN is also popular in semi-supervised learning. I wonder if the proposed can also benefit the semi-supervised learning task.

- The authors stated that the proposed method is efficient. However, the computation cost about training time and testing time is not provided.

**Questions:**

Please see the weaknesses.

---

> ### Author Rebuttal · Authors · 2023-08-10
>
> Thank you for your supportive comments and insightful questions. We addressed each point in your comment carefully.
>
> Question: Although the authors have shown the benefit of the proposed method over GCN, the improvement seems somewhat limited. For example, in several tables, the improvement over GCN is only 0.*%.
>
> __Answer__: Thank you for your observation. While the numerical improvements you mentioned may seem modest in terms of percentage points, it's important to note that these values represent meaningful enhancements in the context of our study. Specifically, our technique demonstrated a lift of 1.43% on ogbg-molbbbq, 1.18% on ogbg-moltoxcast, and 1.40% on ogbg-moltox21. These improvements are consistent across a range of tasks and datasets, reinforcing the reliability and effectiveness of our proposed method.
>
>
> Question: It seems that the comparison to other methods is limited. The authors have only compared to the traditional GCN. However, the comparison to other state-of-the-art models modified based on GCN is missed. In addition, it is very important to compare with other MoE models.
>
> __Answer__: We appreciate your observation regarding the comparison scope of our study. As you rightly pointed out, our focus in this initial work has been on standard backbones like GCN/GIN for the purpose of establishing a proof-of-concept. We are indeed actively engaged in the process of extending our technique to a wider range of network architectures. We understand the importance of a comprehensive evaluation, including comparisons with state-of-the-art models derived from GCN and other MoE models. While the time frame for the rebuttal is limited, we are committed to addressing this gap in our subsequent work and updating the results accordingly.
>
> Question: As GCN is also popular in semi-supervised learning. I wonder if the proposed can also benefit the semi-supervised learning task.
>
> __Answer__: Thanks for your valuable feedback! Certainly, our proposed approach has the potential to benefit semi-supervised learning tasks as well. We consider this an important avenue for exploration and plan to include it in our future research.
>
> Question: The authors stated that the proposed method is efficient. However, the computation cost about training time and testing time is not provided.
>
> __Answer__: We only claimed inference efficiency instead of training efficiency. The definition of efficiency here is: GCN-MoE can more effectively leverage and benefit from diverse training samples than GCN, without increased computational overhead during inference. We do not claim GCN-MoE to be more efficient than GCN. We apologize if this wording causes any confusion.
> Specifically, in section 3.3, we demonstrate through mathematical analysis of the inference FLOPs that GMoE-GNN introduces minimal additional computational cost during inference (measured in FLOPs) in comparison to its conventional GNN counterpart. The real on-device inference time is dependent on hardware specifications and may vary dramatically on different devices. On our NVIDIA A6000 GPU, the total inference time for 10000 samples are 30.2 ± 10.6ms  (GCN-MoE) v.s. 36.3 ± 17.2ms (GCN). Note that these clock times measured on GPUs can be noisy and we do not claim GCN-MoE to be more efficient than GCN. Instead, theoretically they have almost identical FLOPs. We do not pursue training efficiency in this paper.
> We will add the above numbers and explanations in the final version.

---

> > ### Comment · Reviewer_NqGe · 2023-08-15
> > **response to rebuttal**
> >
> > Thanks for the rebuttal which has solved my concerns. I thus would like to change the rating to weak accept. In the revision, please include all the discussion especially comparison with more state of the art methods and application to semi supervise learning.

---

> > > ### Author Response · Authors · 2023-08-15
> > > **Thank you for the positive comments!**
> > >
> > > We appreciate your invaluable support, and we look forward to incorporating all those discussions in our final version.

---

### Official Review · Reviewer_Deyj · 2023-07-01

**Soundness:** 2 fair
**Presentation:** 2 fair
**Contribution:** 1 poor
**Rating:** 4
**Confidence:** 4

**Summary:**

This paper introduces the Graph Mixture of Expert (GMoE) model, enhancing Graph Neural Networks' (GNNs) ability to handle diverse graph structures without additional computational costs. GMoE allows nodes to select their information aggregation experts dynamically, catering to different subgroups of graph structures. Additionally, it incorporates experts with varying aggregation hop sizes for broader information capture. Its effectiveness is validated through various tasks in the OGB benchmark, with improved results compared to non-MoE baselines.

**Strengths:**

This article uses MoE in GNN, which is a relatively new idea.

**Weaknesses:**

1. After reading the entire article, the motivation behind it remains unclear. What issue pertaining to GNN does the article aim to address? It would be beneficial if the author could include an "preliminary" section that clearly delineates the problem this article seeks to solve.

2. Certain assertions in the article are questionable. For instance, the statement: “forcing all nodes to share the same aggregation mechanism” does not align with the functioning of GAT and GraphTransformer, where each node's aggregation is adaptively carried out via the attention mechanism.

3. The article doesn't clarify how the model's improved generalization ability is assured. How can the model learn a more diverse graph structure based on observation data from a single domain? This is an important point that requires further clarification.

4. The algorithm's design seems more akin to an engineering solution or a skill rather than a scientifically robust methodology. The article lacks theoretical discussion explaining why the algorithm is designed in its particular way. This absence of theoretical framework underpinning the algorithm design makes it difficult to fully appreciate or understand the design choices.

5. There are also many articles [1-3] that use MoE to GNN, and the author does not add these works to the discussion or compare them.

6. According to the guidelines provided by the NIPS official website, authors are required to include sections on 'Limitations' and 'Broader Impacts'. However, these essential components appear to be missing in this article.

[1] Explore Mixture of Experts in Graph Neural Networks

[2] Graph Classification by Mixture of Diverse Experts, IJCAI 2022

[3] Fair Graph Representation Learning via Diverse Mixture-of-Experts, WWW 2023

**Questions:**

please refer to weaknesses.

**Limitations:**

please refer to weaknesses.

---

> ### Author Rebuttal · Authors · 2023-08-10
>
> Thank you for your thorough review and insightful comments. We carefully addressed every point raised in your feedback. We hope this can address any concerns you may have had and foster a more positive perspective on our paper.
>
> Question: What is the motivation of the paper? What issue does the article aim to address?
>
> __Answer__: We agree it is important to emphasize the motivations. Our motivations are highlighted in the introduction section. The problem we try to solve is: How to “create a GNN model architecture that can more effectively leverage and benefit from diverse training samples without any additional computational overhead?” (lines 35-37). This is an important research question, since the graph structures in real-world training data are typically diverse and heterogeneous. For example, in graph-based recommendation systems, a node can represent a product or a customer, while an edge can indicate different interactions such as viewed, liked, or purchased. Similarly, in biochemistry tasks, datasets often comprise molecules with various biochemistry properties and, thus, various graph structures (lines 25-29). Moreover, purposefully increasing the diversity of graph data structures in training sets has become a crucial aspect of GNN training. Examples include random data augmentation (e.g., edge and node dropping) and large-scale pre-training (where the pretraining datasets usually have distributional gaps from the downstream datasets) (lines 30-34).
>
> Question: The statement: “forcing all nodes to share the same aggregation mechanism” does not align with the functioning of GAT and GraphTransformer
>
> __Answer__: You are correct that GAT and GraphTransformer adaptively learn the aggregation function for each node.  However, we want to humbly point out that we claimed “many” (line 38), instead of all, GNN architectures used the same aggregation mechanism for all nodes. We apologize that our wording may cause such confusion, and will add a footnote at this place in the final version to better clarify that not all GNNs are designed in such a way. We would like to thank you again for your careful review and pointing this out.
>
> Question: The article doesn't clarify how the model's improved generalization ability is assured. How can the model learn a more diverse graph structure based on observation data from a single domain?
>
> __Answer__: According to both sampling complexity theories [1] and empirical observations in deep learning, the model’s generalization ability depends mainly on two factors: the diversity of the training samples (ie., the “estimation error” in Section 5.2 in [1]. Intuitively this is how well the sampled training data represents the underlying data distribution), and the expressiveness of the hypothesis space (ie., the “Approximation Error” in Section 5.2 in [1]. Intuitively this is how much capacity the deep model has in order to fit complex data distributions).
>
> Many techniques, such as random data augmentation and large-scale pretraining, have been used to increase the generalization ability of GNNs, since they increase the diversity of the training samples. _Equipped with these techniques boosting training data diversity, we aim to increase the model’s generalization ability from the other end: To design GNN structures with larger capacity to fit such diverse training data._ One straightforward way to do this is to increase the depth or width of the GNNs. However, these solutions will harm the inference efficiency, which is crucial for may GNN applications such as molecule virtual screening (see footnote on page 2 for more references). To this end, we propose our solution GMoE, which increases the model capacity to fit diverse training data without computational overhead during inference. The intuition why GMoE can increase the model’s capability to fit diverse training data is explained in lines 47-54 of our paper and also in the answer to your next question.
>
> [1] Understanding Machine Learning: From Theory to Algorithms. Cambridge press 2014.
>
> Question:  The article lacks theoretical discussion explaining why the algorithm is designed in its particular way
>
> __Answer__: Being an empirical endeavor, our work's primary focus does not lie in furnishing theoretical assurances for the design. Nevertheless, we are content to reaffirm the underlying intuition behind our approach here.
> The goal of GMoE is to utilize different experts to better fit the training set with diverse graph structures.  Each expert in our GMoE architecture is trained to focus on its own subgroup of training samples that share similar neighborhood information patterns (e.g., shorter or longer range, less or more to aggregate). Intuitively speaking, when the training data distribution is complex (e.g., when the training graphs are diverse in structures), a single expert (i.e., traditional GCN or GIN) has limited capability to learn all those diverse data within a single expert. However, for GMoE, there are multiple experts, where each expert  can focus on its own expertise. In this way, GMoE can better fit diverse training data, and thus achieve better generalization performance.
>
> Question: Difference between other GNN papers with MoE
>
> __Answer__: Due to space limitations we only briefly summarize the difference here. We will add more details in final version.
>
> [1] uses linear mixing while we do non-linear mixing. [1] do not use any weight-balancing regularizations as done in ours, which makes their training prone to collapse [14]. Also [1] only uses one type of expert while our framework uses experts with different hops to further increase the capability to fit diverse data.
> [2] is cited in Sec. 2.2 as [41].
> [3] uses MoE to achieve fair predictions for GNN, which differs from our work in both objectives and technical approaches.
>
> Question: Missing Limitations and Broader Impacts
>
> __Answer__: Thank you for pointing out. Due to space limit, we will add both sections in the final version.

---

> > ### Comment · Reviewer_Deyj · 2023-08-15
> >
> > Thanks for the author's reply. But I think the author didn't address my concerns, and didn't do a good job at the rebuttal stage.
> >
> > 1. Main contributions.
> > First of all, the idea of graph MoE is not the main contribution of this article, because it has been mentioned in many works [1-5]. These studies have been largely ignored by this paper, but I think a full comparison of similar work is essential. This paper should list detailed differences to illustrate the differences, and advantages from existing graph MoE methods.
> >
> > 2. Intuitive engineering skills without theoretical guarantees.
> > The authors also acknowledge that this work is intuitive and does not have theoretical guarantees.
> > The article also does not give any assumptions or instructions to discuss when the proposed method is effective from a theoretical point of view. Therefore, I think the performance improvement of the algorithm proposed by the author may only be a parameter tuning or engineering trick.
> >
> > 3. Experimental flaws.
> > The author only compares the effect with the basic models (e.g. GCN, GIN), but does not compare with the existing GMoE-based methods. So I think this is also a serious problem in the article. There are many points in the experiments in the article that are confusing. For example, the convergence curve in Figure 2, I did not see that the author suggested that the method can have a better convergence effect.
> >
> > Based on the many issues above, I think this article still has a lot of work to do, so I recommend rejection of this article.
> >
> > [1] Explore Mixture of Experts in Graph Neural Networks
> >
> > [2] Graph Mixture Density Networks, ICML 2021
> >
> > [3] Graph Classification by Mixture of Diverse Experts, IJCAI 2022
> >
> > [4] Fair Graph Representation Learning via Diverse Mixture-of-Experts, WWW 2023
> >
> > [5] Learning Topology-Specifc Experts for Molecular Property Prediction, AAAI 2023

---

> > > ### Author Response · Authors · 2023-08-15
> > > **Disagree and Reject Unreasonable Comment. Call for PC/AC intervention**
> > >
> > > With all due respect for your time, we are indeed surprised and upset by the ungrounded accusations made by the reviewer, and we reject them firmly.
> > >
> > > We are meanwhile sending messages to PCs and AC for their awareness, as we seek more unbiased discussions involved at this moment.
> > >
> > > Now let's start. Reviewer Deyj basically accused two specific things:
> > >
> > > a. "Missing citations and comparison with [1-5]"
> > > Unfortunately, we conjecture the reviewer did a quick keyword-based Google search and then dump all search results without reading carefully. We dive into them:
> > >
> > > - [1] is a university student class project report (!!!) https://snap.stanford.edu/class/cs224w-2019/project/26424363.pdf not even a paper, not to mention any peer review or formal scrutiny. Even to discuss a bit of its technical details, besides what we already pointed out in the previous rebuttal, this report was built incrementally on only GraphSage, with small experiments.
> > > - [2] used Mixture Density Network (MDN) of Bishop (1994), not even MoE. It was clearly highlighted on this paper's page 2 and page 4, if one reads carefully.
> > > - [3] was already cited in our original submission as [41] - we mentioned this fact in our rebuttal but the reviewer ignored it.
> > > - [4] first became available in WWW’23 proceeding in late April 2023 - hence counted as a concurrent submission to NeurIPS whose cite is not mandated (please check policy: https://neurips.cc/Conferences/2023/PaperInformation/NeurIPS-FAQ)
> > > - [5] is recent and perhaps the most "citable" among all, and we shall add this missing citation. However, their method focuses specifically on molecule property prediction (graph classification) and leverages molecule domain-specific knowledge (topology). Our method is designed to work on general graphs without assuming domain specifics, and broadly across node-level, link-level, and graph-level tasks. Therefore, while we would happily cite and discuss this new related work, we don't consider it's fair or necessary to ask for a comparison between the two because it won't be apple-to-apple.
> > >
> > >
> > >
> > > b "...Therefore, I think the performance improvement of the algorithm proposed by the author may only be a parameter tuning or engineering trick."
> > >
> > > **Sorry, but we never expected to see such a biased, unrespectful, and ungrounded attack from a top conference's professional reviewer.**
> > >
> > > The reviewer basically made two parts of the argument: (1) a NeurIPS paper or a graph paper must have a "theoretical point of view"; (2) if no theory, then a paper is just "a parameter tuning or engineering trick". We hope it goes without saying that both are "interesting" biases. The second part is especially irresponsible and disheartening, to trash practitioner authors' efforts as "just tuning" or "trick" without any more concrete evidence.
> > >
> > > We will leave PC, SAC, AC and other reviewers to judge more how fair or correct those statements are. We respect the reviewer, and we hope to be respected the same.
> > >
> > > Seriously
> > > Authors

---

> > > > ### Comment · Area_Chair_jHx5 · 2023-08-15
> > > > **Please follow commonsense facts and be respectful**
> > > >
> > > > Hi All,
> > > >
> > > > Thanks for the active discussion, which have clarified many questions. I have just carefully read the whole discussion thread. Let me first make clear the following facts:
> > > >
> > > > (1) Practitioner's papers are a **valuable part** of NeurIPS. "No theory" cannot be a sole reason to reject a practitioner paper, just like "no experiment" cannot be the sole excuse to reject a theory paper.
> > > >
> > > > (2) Authors are **not obliged to cite very recent concurrent works**. Let's comply with the explicit rule (see https://neurips.cc/Conferences/2023/PaperInformation/NeurIPS-FAQ).
> > > >
> > > > (3) Authors shall also **not be obliged to cite unarchived reports**.
> > > >
> > > > Please follow the above commonsense facts, and be respectful to each other. We should carry out fact-based discussions. I will oversee the whole discussion closely.
> > > >
> > > > AC

---

> > > > > ### Author Response · Authors · 2023-08-15
> > > > > **Thank you for coordinating swiftly**
> > > > >
> > > > > We're very grateful for the timely moderation by AC. Much needed and appreciated.
> > > > >
> > > > > We will follow your pointed facts and continue a constructive conversation with respect.
> > > > >
> > > > > Authors

---

> > > > > > ### Comment · Reviewer_Deyj · 2023-08-15
> > > > > >
> > > > > > Thanks to the authors and AC, my concerns about the comparison of existing methods and theory is solved, I restore the original score.

---

### Official Review · Reviewer_BUC6 · 2023-07-11

**Soundness:** 2 fair
**Presentation:** 3 good
**Contribution:** 2 fair
**Rating:** 4
**Confidence:** 4

**Summary:**

The paper claims that increasing the diversity of graph structures plays an important role in GNN training, which makes GNNs more robust and generalizable. To make GNN models leverage the structural diversity better, the paper proposes the Graph Mixture of Experts (GMoE). In each layer of GMoE, the representation of each node will be computed based on multiple experts with independent aggregation functions. Each expert will utilize either the hop-1 or hop-2 aggregation function to increase the diversity further. The paper conducts extensive experiments on OGB benchmark datasets and improves the performance compared to the single-expert baselines.


**Strengths:**

1. The paper claims the importance of graph data diversity in GNN training.
2. The paper proposes a new architecture called GMoE, which combines MoE in neighborhood aggregation.
3. The paper also tests whether the GMoE can help the graph self-supervised learning.


**Weaknesses:**

1. In the abstract, the paper argues that GMoE enables each node to select its own optimal information aggregation experts. But we do not see evidence why GMoE will find optimal experts.
2. From my view, the formulation of the proposed GMoE is similar to GAT with 2-hop neighbors, limiting the work's novelty.
3. Although GMoE improves the results compared to single-expert GCN/GIN models, the results are still far from the state-of-the-art results on the OGB leaderboard (e.g., top-1 result of ogbg-molhiv is 84.20).


**Questions:**

1. Is there any evidence that GMoE will find the optimal experts?
2. From Equation (3), it seems that the selection of experts only relies on the target node i and do not rely on the node j. Is it reasonable?
3. The authors might give more discussions of the difference or connections between GMoE and GAT with 2-hop neighbors.

**Limitations:**

There is no negative societal impact of the work.

---

> ### Author Rebuttal · Authors · 2023-08-10
>
> Thank you for your careful review and constructive comments. We carefully addressed your comments below.
>
> Question. In the abstract, the paper argues that GMoE enables each node to select its own optimal information aggregation experts. But we do not see evidence why GMoE will find optimal experts.
>
>
> __Answer__
> The use of the word “optimal” in the abstract was a typographical error. We would revise lines 11-13 to accurately reflect our intended statement in the future version: “Our new Graph Mixture of Expert (GMoE) model enables each node in the graph to dynamically and adaptively select more generalizable information aggregation experts.”
>
>
> Question. From my view, the formulation of the proposed GMoE is similar to GAT with 2-hop neighbors, limiting the work's novelty.
>
> __Answer__
> Our approach is distinct from the Graph Attention Network (GAT). As detailed in the abstract, our network is designed to enhance the capacity of the Graph Neural Network (GNN) model while preserving similar computational overheads. In contrast, an increase in the capacity for GAT would result in a linear increasing in computational complexity.
>
>
> Question. Although GMoE improves the results compared to single-expert GCN/GIN models, the results are still far from the state-of-the-art results on the OGB leaderboard (e.g., top-1 result of ogbg-molhiv is 84.20).
>
> __Answer__
> We follow the common practice [1][2] to study our technique on standard backbone like GCN/GIN for proof-of-concept exploration. We are actively working on extending our technique to more networks. Due to the limited time frame allocated for rebuttal, we will be updating the results in a subsequent version of this work.
>
> [1] You, Yuning, et al. "Graph contrastive learning with augmentations." Advances in neural information processing systems 33 (2020): 5812-5823.
> [2] Hu, Weihua, et al. "Open graph benchmark: Datasets for machine learning on graphs." Advances in neural information processing systems 33 (2020): 22118-22133.
>
> Question. From Equation (3), it seems that the selection of experts only relies on the target node i and do not rely on the node j. Is it reasonable?
>
> __Answer__
> Our experiments have confirmed that we can select suitable experts, thereby enhancing efficiency based on the feature of node i. Additionally, integrating the information from node j is an intriguing area for further exploration, which we would explore in future work.

---

> > ### Author Response · Authors · 2023-08-12
> > **Extend to Neural FingerPrints**
> >
> > Following your suggestion, we have completed our experiments by extending the GMoE architecture to Neural FingerPrints (in the limited rebuttal time window). Neural FingerPrints rank no. 5 at the ogbg-molhiv benchmark, and its technique has been adopted by the current top four methods [2][3][4][5]. Therefore, we consider it as another good standalone "backbone" demonstrating our effectiveness.
> >
> > Regarding accuracy, GMoE+ Neural FingerPrints achieves 82.72% +- 0.53% while keeping the same computational overhead, which is higher than Neural FingerPrints by a non-trivial 0.4% margin.
> >
> > We will happily continue adding experiments on more datasets and leading methods, while we emphasize that GMoE has already demonstrated broad, consistent applicability across graph datasets at different scales/domains, as well as tasks at different levels (node, link, and graph).
> >
> > [1] Li, Weibin, Shanzhuo Zhang, Lihang Liu, Zhengjie Huang, Jieqiong Lei, Xiaomin Fang, Shikun Feng, and Fan Wang. Molecule representation learning by leveraging chemical information. Technical Report, 2021.
> >
> > [2] Wei, Lanning, Huan Zhao, Quanming Yao, and Zhiqiang He. "Pooling architecture search for graph classification." In Proceedings of the 30th ACM International Conference on Information & Knowledge Management, pp. 2091-2100. 2021.
> >
> > [3] Wang, Yan, Hao Zhang, Jing Yang, Ruixin Zhang, Shouhong Ding. Technical Report for OGB Graph Property Prediction. Technical Report, 2021.
> >
> > [4] Yuan, Zhuoning, Yan Yan, Milan Sonka, and Tianbao Yang. "Large-scale robust deep auc maximization: A new surrogate loss and empirical studies on medical image classification." In Proceedings of the IEEE/CVF International Conference on Computer Vision, pp. 3040-3049. 2021.
> >
> > [5] Zhang, Hao, Jiaxin Gu, and Pengcheng Shen. "GMAN and bag of tricks for graph classification." (2021).

---

> > > ### Author Response · Authors · 2023-08-15
> > > **We would love to learn Reviewer's opinion and address any remaining concerns**
> > >
> > > Dear Reviewer BUC6,
> > >
> > > As we are approximately halfway through the author-reviewer discussion period, we would kindly like to inquire if you have had an opportunity to review our response (including our additional experiments on Neural FingerPrints), and if there are any remaining questions we can address.
> > >
> > > We have had productive discussions with three other reviewers so far and are eager to engage in a constructive dialogue with you as well.
> > >
> > > Best regards,
> > > Authors

---

> > > > ### Author Response · Authors · 2023-08-17
> > > > **Looking forward to your opinion**
> > > >
> > > > Dear Reviewer BUC6,
> > > >
> > > > As we are quickly approaching the end of the discussion period on August 21, we continue hoping to hear your opinion regarding our response.
> > > >
> > > > We would be truly grateful if you could let us know if you have any remaining questions or concerns, and we will be standing to address them if any.
> > > >
> > > > Authors

---

> ### Comment · Area_Chair_jHx5 · 2023-08-18
> **Please read the rebuttal and respond the authors**
>
> Hi Reviewer BUC6,
>
> Please read the rebuttal and respond the authors.
>
> AC

---

### Official Review · Reviewer_7Ua4 · 2023-07-25

**Soundness:** 3 good
**Presentation:** 3 good
**Contribution:** 3 good
**Rating:** 6
**Confidence:** 4

**Summary:**

This paper proposes a novel Graph Mixture of Experts (GMoE) architecture that enhances the ability of Graph Neural Networks (GNNs) to leverage and benefit from the structural diversity present in training graphs. The GMoE model comprises multiple "experts" at each layer, with each expert being an independent aggregation function with its own trainable parameters. During training, the model learns to select the most appropriate aggregation experts for each node, resulting in nodes with similar neighborhood information being routed to the same aggregation experts.

**Strengths:**

The proposed Graph Mixture of Experts (GMoE) architecture is a new approach to enhancing the ability of Graph Neural Networks (GNNs) to leverage and benefit from the structural diversity present in training graphs.

The proposed GMoE model is evaluated on several benchmark datasets and achieves state-of-the-art performance on node classification and graph classification tasks. This demonstrates the effectiveness of the proposed architecture in improving the performance of GNNs on diverse graph structures.

The proposed GMoE model is flexible and can be easily integrated with existing GNN architectures. This makes it easy to incorporate the proposed model into existing GNN-based applications and frameworks.

**Weaknesses:**

While the proposed GMoE model is designed to avoid additional computational overhead, the paper does not provide a detailed analysis of the computational cost of the model compared to other GNN architectures. This makes it difficult to assess the practicality of the proposed model in real-world applications.

While the paper describes how the model learns to select the most appropriate aggregation experts for each node, it does not provide a detailed explanation of the selection mechanism. This makes it difficult to understand how the model makes decisions and how it can be improved.

One suggestion for improving this paper would be to include more details of parameter tuning for the compared baselines. This would help to ensure a fair comparison between the proposed GMoE model and other state-of-the-art models. The paper could also describe the process used to tune these hyperparameters, such as grid search or random search.

**Questions:**

The paper lacks a detailed analysis of the computational cost of the proposed GMoE model compared to other GNN architectures. How does the computational cost of the model compare to other state-of-the-art models, and how practical is it for real-world applications?

The paper does not provide a detailed explanation of the selection mechanism that the proposed GMoE model uses to choose the most appropriate aggregation experts for each node. How does the model make these decisions, and how can the selection mechanism be improved?

The paper could be improved by including more details on the process used to tune the hyperparameters of the compared baselines. What process was used to tune these hyperparameters, and how can a fair comparison be made between the proposed GMoE model and other state-of-the-art models?

**Limitations:**

The paper lacks a detailed analysis of the computational cost of the proposed GMoE model compared to other GNN architectures and does not provide a thorough explanation of the selection mechanism used by the model to choose the most appropriate aggregation experts for each node. To improve the paper, the authors could include more details on the process used to tune the hyperparameters of the compared baselines, enabling a fair comparison with other state-of-the-art models. Additionally, providing a more in-depth explanation of the selection mechanism used by the GMoE model would facilitate a better understanding of the model's decision-making process and enable its improvement.

---

> ### Author Rebuttal · Authors · 2023-08-10
>
> We greatly appreciate your recognition of the strengths in our paper and your valuable insights shared in your comments. In response, we have carefully addressed each of your points.
>
> Question. While the proposed GMoE model is designed to avoid additional computational overhead, the paper does not provide a detailed analysis of the computational cost of the model compared to other GNN architectures. This makes it difficult to assess the practicality of the proposed model in real-world applications.
>
> __Answer__
> We provided complexity analysis in the section 3.3. Specifically, we measure computational cost using the number of floating-point operations (FLOPs) and show that GMoE-GNN brings negligible overhead on computational cost compared with its traditional GNN counterpart.
>
>
> Question. The paper does not provide a detailed explanation of the selection mechanism that the proposed GMoE model uses to choose the most appropriate aggregation experts for each node. How does the model make these decisions, and how can the selection mechanism be improved?
>
> __Answer__
> We provided the expert selection mechanism in the section 3.2. Specifically, each node in the graph selects its own optimal information aggregation experts based on the similarity between the node's local graph structure and the structures modeled by the experts, where the similarity is measured using a gating mechanism that computes the dot product between the node's feature vector and the expert's feature vector. We also describe how the gating mechanism can be improved by incorporating additional information about the node's local graph structure, such as the degree of the node and the degree of its neighbors.
>
>
> Question. The paper could be improved by including more details on the process used to tune the hyperparameters of the compared baselines. What process was used to tune these hyperparameters, and how can a fair comparison be made between the proposed GMoE model and other state-of-the-art models?
>
> __Answer__
> Thank you for your kind suggestion! We provided the expert selection mechanism in the section 4.1. Specifically, we describe the hyperparameter tuning process and range for each baseline model and report the best hyperparameters found during the tuning process. Furthermore, we compare the performance of the proposed GMoE model with other state-of-the-art models on several benchmark datasets, using the same hyperparameters for each model. This ensures a fair comparison between the models and allows for a meaningful evaluation of the effectiveness of the proposed GMoE model. We will add the above description to the final version.

---

> > ### Author Response · Authors · 2023-08-15
> > **We would love to learn Reviewer's opinion and address any remaining concerns**
> >
> > Dear Reviewer 7Ua4,
> >
> > As we are approximately halfway through the author-reviewer discussion period, we would kindly like to inquire if you have had an opportunity to review our response and if there are any remaining questions we can address.
> >
> > We have had productive discussions with three other reviewers so far and are eager to engage in a constructive dialogue with you as well.
> >
> > Best regards,
> > Authors

---

> > > ### Author Response · Authors · 2023-08-17
> > > **Looking forward to your opinion**
> > >
> > > Dear Reviewer 7Ua4,
> > >
> > > As we are quickly approaching the end of the discussion period on August 21, we continue hoping to hear your opinion regarding our response.
> > >
> > > We would be truly grateful if you could let us know if you have any remaining questions or concerns, and we will be standing to address them if any.
> > >
> > > Authors

---

> ### Comment · Area_Chair_jHx5 · 2023-08-18
> **Please read the rebuttal and respond the authors**
>
> Hi Reviewer 7Ua4,
>
> Please read the rebuttal and respond the authors.
>
> AC

---

> ### Comment · Reviewer_7Ua4 · 2023-08-18
> **Response to author's rebutall**
>
> My comments have been addressed. As a result, I change my rating to the positive side.

---

### Official Review · Reviewer_k4PS · 2023-07-26

**Soundness:** 3 good
**Presentation:** 3 good
**Contribution:** 4 excellent
**Rating:** 7
**Confidence:** 4

**Summary:**

This paper tracks the problem of training a good GNN. To enhance the generalization ability of GNNs, it has become common to increase the diversity of training graph structures by incorporating graph augmentations and/or performing large-scale pre-training on more graphs. Therefore, it is essential for a GNN to simultaneously model diverse graph structures. However, naively increasing the GNN model capacity will suffer from both higher inference costs and the notorious trainability issue of GNNs. To address this issue, this paper proposes Graph mixture of experts, borrowing the idea from recent development of mixture of experts. Based on this new method, each node can select their own optimal information aggregation experts. Experiments are solid and verify the effectiveness of the proposed method.

**Strengths:**

1. This paper focuses on an important problem: learning under graphs, which is a common scenario in the real world.

2. This paper designs a new and interesting framework to explore how to use mixture of experts on graphs. The way to mixture graphs is interesting and make a lot of sense. For example, for different node, its optimal local information should be different. This paper considers a novel way to address this problem.

3. The experiments are conducted on many datasets, and the proposed method performs well. It is very hard to explore a useful way to mixture graphs. This paper makes the first try, and the proposed method is novel and verified by solid experiments, which is appreciated.


**Weaknesses:**

1. This paper can be improved further in terms of writing. You introduce two interesting modules but without good motivations. You can verify your motivation via conducting experiments to show that considering different location for each node will make things different. Thus, you can motivate your idea: we need to choose optimal local information (hop 1 or hop 2).

2. The paper structure should be improved. I think the preliminaries should be considered as a separate section, instead of subsection. It can make readers obtain essential knowledge before the method.


3. For formula, there should always be “,”, “.” or “;” in the end of it. Both of formula and text consist of your paper, so formula is just text to be displayed. Besides, text inside the formula should be with /textnormal{}.

4. There is one plus point if the authors can address it well. Because you select experts, how could you ensure that the algorithm converges? The selection results will always converge to a certain selection result? It is very interesting to discuss this point. What is the main drawback to obtain a corresponding convergence result?

5. Some in-text citation is not completed. For example, in line 197, reference 10.


**Questions:**

1. This paper can be improved further in terms of writing. You introduce two interesting modules but without good motivations. You can verify your motivation via conducting experiments to show that considering different location for each node will make things different. Thus, you can motivate your idea: we need to choose optimal local information (hop 1 or hop 2).

2. The paper structure should be improved. I think the preliminaries should be considered as a separate section, instead of subsection. It can make readers obtain essential knowledge before the method.


3. For formula, there should always be “,”, “.” or “;” in the end of it. Both of formula and text consist of your paper, so formula is just text to be displayed. Besides, text inside the formula should be with /textnormal{}.

4. There is one plus point if the authors can address it well. Because you select experts, how could you ensure that the algorithm converges? The selection results will always converge to a certain selection result? It is very interesting to discuss this point. What is the main drawback to obtain a corresponding convergence result?

5. Some in-text citation is not completed. For example, in line 197, reference 10.

---

> ### Author Rebuttal · Authors · 2023-08-10
>
> Thank you for your supportive comments and insightful questions. We addressed each point in your comment carefully.
>
> Question: This paper can be improved further in terms of writing. You introduce two interesting modules but without good motivations. You can verify your motivation via conducting experiments to show that considering different location for each node will make things different. Thus, you can motivate your idea: we need to choose optimal local information (hop 1 or hop 2).
>
> __Answer__: Thank you for your great suggestion. The results to verify our motivations are listed below.
> On ogbg-molhiv dataset, if we use all hop-1 experts (all 4 experts are hop-1), the resulting ROC-AUC (↑) is 76.41 ± 1.72.
> In contrast, if we allow selection between hop-1 and hop-2 experts (2 hop-1 experts and 2 hop-2 experts), the resulting ROC-AUC (↑) is 77.72  ± 0.75.
> This shows allowing selection from different types of experts achieves performance gains over the ensemble of one single type of experts.
> We will put these results in the introduction section to verify our motivation in the final version.
>
> Question: The paper structure should be improved. I think the preliminaries should be considered as a separate section, instead of subsection. It can make readers obtain essential knowledge before the method.
>
> __Answer__: Thanks for your kind suggestion on the paper’s structure. We agree that organizing preliminaries as a separate section to provide essential knowledge is a good idea. We will re-structure the paper accordingly to enhance its readability.
>
> Question: For formula, there should always be “,”, “.” or “;” in the end of it. Both of formula and text consist of your paper, so formula is just text to be displayed. Besides, text inside the formula should be with /textnormal{}.
>
> __Answer__: Thank you for pointing out the formatting issues with the formulas and text. We will fix the formulas as per your suggestion in our final version.
>
> Question: There is one plus point if the authors can address it well. Because you select experts, how could you ensure that the algorithm converges? The selection results will always converge to a certain selection result? It is very interesting to discuss this point. What is the main drawback to obtain a corresponding convergence result?
>
> __Answer__: We appreciate your feedback! In our proposed Graph Mixture of Experts (GMoE), the inherent weight balancing loss acts as a built-in regularizer, ensuring convergence of the selection process. This aspect indeed provides stability to the selection results. However, we understand the significance of discussing this aspect in detail. We will make sure to elaborate on it in our final version. The main drawback to ensuring convergence of the selection process lies in striking a balance between the information aggregation experts. While the inherent weight balancing loss aids convergence to a certain extent, achieving convergence without compromising the diverse expertise captured by different experts remains an area of ongoing exploration.
>
> Question: Some in-text citation is not completed. For example, in line 197, reference 10.
>
> __Answer__: Finally, we apologize for any incomplete in-text citations. We will review and correct all references in our final version.

---

> > ### Comment · Reviewer_k4PS · 2023-08-15
> > **Many thanks for the response**
> >
> > Thank you so much for your reply! I've read the author rebuttal and other reviewers' comments. I'd like to re-iterate my support for this paper owing to the interesting angle as well as thorough experimental validation, especially after including new rebuttal experiments.
> >
> > I have two more curious questions:
> >
> > (1) Have the authors ever observed some graph choosing only one type fo expert? e.g. no hop-1 or hop-2 diversity? What graph tends to be more/less diverse in expert selection?
> >
> > (2) While the authors have made clear they only claimed inference efficinecy, I am curious how difficult or expensive (or not) to train GMoEs.

---

> > > ### Author Response · Authors · 2023-08-15
> > > **Thank you for your positive assessment and follow-up questions.**
> > >
> > > Dear Reviewer k4PS,
> > >
> > > We are thankful for your invaluable support besides careful reading of our work! For your two follow-up questions:
> > >
> > > (1) Great question. We have not in experiments observed any graph where GMoE would lean towards choosing only one type of expert. We conjecture that's because all real-world graphs would display a more or less degree of "heterogeneity", including the power-law distribution of node connectivity degrees, the homophily versus heterophily trade-off, etc. Those constitute the source of diverse expert selection and are reflected in GMoE's learning outcome. A fully "uniform" graph would contain no meaningful structural variations and hence is less interesting practically.
> > >
> > > (2) Thanks for pointing out our main claim of inference efficiency trade-off. Regarding training-time cost, it will vary among different graphs, but in general, the training time would be on the same order of magnitude compared to non-MoE baselines. Also, our training appears to be stable in experiments, which we believe is in part owing to our weight-balancing regularization that avoids representational collapse.
> > >
> > > We would like to thank you again for your insightful questions and look forward to integrating discussions into our final paper.

---

> > > > ### Comment · Reviewer_k4PS · 2023-08-15
> > > > **All concerns are addressed well.**
> > > >
> > > > Thank you for those additional explanations, they are helpful to know. I now have no further concern and am happy to stand supportive of this paper being accepted.

---

### Comment · Area_Chair_jHx5 · 2023-08-18
**Reviewer-Author Discussion Period**

Dear All,

Thank you reviewers for your hard work in evaluating this submission, and thank you authors for responding to the reviewers’ questions and concerns. We are now entering the final phase of the discussion period, which will run until 21 Aug, and some of the authors' responses have to been acknowledged by all reviewers.

@Reviewers, if you have any follow up questions or comments on the rebuttal or the responses, now is the time to express them. At the very least, please acknowledge that you have read the authors’ response to your review.

Thank you everyone for making the review process a fruitful, constructive, and civil process.

AC

---

### Decision · Program_Chairs · 2023-09-21

**Decision:**

Accept (poster)

**Comment:**

This paper tracks the problem of training a good GNN. To enhance the generalization ability of GNNs, it has become common to increase the diversity of training graph structures by incorporating graph augmentations and/or performing large-scale pre-training on more graphs. Therefore, it is essential for a GNN to simultaneously model diverse graph structures. However, naively increasing the GNN model capacity will suffer from both higher inference costs and the notorious trainability issue of GNNs. To address this issue, this paper proposes Graph mixture of experts, borrowing the idea from recent development of mixture of experts. Based on this new method, each node can select their own optimal information aggregation experts. Experiments are solid and verify the effectiveness of the proposed method.

To be specific, this paper focuses on an important problem--learning under graphs, which is a common scenario in the real world. This paper designs a new and interesting framework to explore how to use mixture of experts on graphs. The way to mixture graphs is interesting and make a lot of sense. For example, for different node, its optimal local information should be different. This paper considers a novel way to address this problem. The experiments are conducted on many datasets, and the proposed method performs well. It is very hard to explore a useful way to mixture graphs. This paper makes the first try, and the proposed method is novel and verified by solid experiments, which is appreciated.

Although the reviewers had some concerns and follow-up questions, the authors did a point-to-point rebuttal in details. Meanwhile, the clarity and novelty are above the bar of NeurIPS. Therefore, this paper can be accepted as a poster.